nanotechnology/environmental chemistry

terbium, $TiO_2$ NPs, sol–gel method, photocatalytic mechanism, dye wastewater

**Author for correspondence:**
Shaobai Wen
e-mail: hyhk0898@126.com

# Rapid preparation of terbium-doped titanium dioxide nanoparticles and their enhanced photocatalytic performance

Zhencui Wang[1,2], Yuechao Song[2], Xingfei Cai[3], Jun Zhang[2], Tianle Tang[2] and Shaobai Wen[1,2]

[1]Key Laboratory of Tropical Translational Medicine of Ministry of Education and School of Tropical Medicine and Laboratory Medicine, Hainan Medical University, Haikou, Hainan 571199, People's Republic of China
[2]Laboratory of Environmental Monitoring, College of Tropical and Laboratory Medicine, Hainan Medical University, Haikou 571101, People's Republic of China
[3]Guangdong Tianyuan Environment Co., Ltd, Shenzhen 578061, People's Republic of China

(iD) SW, 0000-0001-6728-6283

Further applications of photocatalysis were limited by the high recombination probability of photo-induced electron–hole pairs in traditional titanium dioxide nanoparticles ($TiO_2$ NPs). Herein, we modified them with rare earth metal via a facile sol–gel method, using tetrabutyl titanate as a precursor and terbium (III) nitrate hexahydrate as terbium (Tb) source. The resulting samples with different Tb doping amounts (from 0 to 2%) have been characterized by X-ray diffraction, UV–visible diffuse reflectance spectroscopy, X-ray photo-electron spectroscopy and a scanning electron microscope. The photocatalytic performance of Tb-doped $TiO_2$ was evaluated by the degradation of methylene blue. The effects of Tb doping amount and initial pH value of solution were investigated in detail. The composite with Tb doping amount of 1.0 wt% showed the highest photocatalytic performance. It exhibited approximately three times enhancement in photocatalytic activity with a reaction rate constant of $0.2314 \, h^{-1}$ when compared with that of commercial P25 ($0.0827 \, h^{-1}$). In addition, it presented low toxicity on zebrafishes with 96 h-$LC_{50}$ of $23.2 \, mg \, l^{-1}$, and has been proved to be reusable for at least four cycles without significant loss of photocatalytic activity. A probable photocatalytic mechanism of Tb-doped $TiO_2$ was proposed according to the active species trapping experiments. The high photocatalytic performance, excellent reusability and low toxicity of Tb-doped $TiO_2$ indicated that it is a promising candidate material in the future treatment of dye wastewater.

# 1. Introduction

With the acceleration of industrialization, water pollution has become one of the most significant problems in society, especially the organic contaminants [1]. The carcinogenic and teratogenic characteristics of organic contaminants have seriously endangered human health. Moreover, the organic pollutants could consume a large amount of dissolved oxygen, which affects the normal growth of aquatic organisms and leads to water deterioration [2,3]. As one of the new advanced oxidation processes, photocatalysis has attracted more attention in recent years because of its remarkable advantages, including the simple operation, high efficiency and no secondary pollution. It has been considered to be the most promising environmentally-friendly technology for wastewater treatment [4,5].

As we know, catalysts play an important role in the photocatalytic process. Among the numerous photocatalysts, titanium dioxide is of particular interest owing to its non-toxicity, unique physical properties and excellent chemical stability. It is widely used in the fields of wastewater treatment, air purification, energy utilization and hydrogen production [6]. The semiconductor titanium dioxide acts as a medium under the excitation of the light, resulting in the migration of electrons from the valence band to the conduction band, the holes ($h^+$) are left behind in the valence band, thus the electrons and holes shift to the semiconductor surface and participate in a series of reduction and oxidation reactions [7]. Accordingly, organic contaminants are decomposed by the reactive groups produced from titanium dioxide. Although the research studies on titanium dioxide have made a great breakthrough, the high recombination probability of photo-induced electron–hole pairs leads to difficulties in its further practical applications [8]. Effectively suppressing the recombination of photo-induced electron–hole pairs is crucial to achieve good photocatalytic activity of the photocatalyst [9]. Therefore, it is indispensable and urgent to modify the traditional titanium dioxide for improving the photocatalytic activity.

Researchers have attempted to fabricate titanium-based composites with precious metals, transition metals, non-metallic elements and metal ions, and rare earth metal ion doping has proved to be an effective means of increasing photocatalytic activity [10–15]. To date, among all rare earth elements used for doping, La is being studied most frequently, followed by Ce, Er, Pr, Gd, Nd, Sm, Ho, etc. while research on increasing photocatalytic performance by doping with Tb and its specific photocatalytic mechanism is rarely reported [16]. The electronic configuration of Tb has a partially filled 4f shell and a 4f electron spin–orbit coupling. Besides, the electron energy of 4f, 5d and 6s is similar, thus it has the capacity to produce more energy levels [17]. Tb doping facilitates the formation of more doping levels, so that electrons trapped on the doping level can be excited by photons with small energy more easily, thereby photon utilization was improved [18]. Furthermore, Tb is a kind of rare earth element with a variable valence, and diffusion of the oxidized or reduced state into the $TiO_2$ lattice causes lattice distortion and expansion, which facilitates the formation of vacancy oxygen (V-O) to improve photocatalytic activity [19]. The preparation methods of titanium-based photocatalytic materials can be mainly divided into physical methods, chemical methods and comprehensive methods. The comprehensive methods include hydrothermal synthesis, water-in-oil microemulsion, $TiCl_4$ gas phase process and titanium alkoxide hydrolysis [20–24]. However, these methods usually have a complicated process or need an expensive apparatus [25]. To our knowledge, the sol–gel step is a rapid process with simple operation, and the crystal size of the resulting product is controllable.

In this study, we fabricated Tb-doped titanium dioxide composite via a facile sol–gel method, and its photocatalytic performance was evaluated by the degradation of methylene blue (MB). Furthermore, the stability and biosafety of the composites were investigated. Most importantly, the corresponding photocatalytic mechanism was further discussed.

# 2. Experimental section

## 2.1. Chemicals and reagents

MB ($C_{37}H_{27}N_3O_9S_3 \cdot Na_2$, greater than 99.7%) and terbium(III) nitrate hexahydrate were purchased from Shanghai Macklin Biochemical Technology Co., Ltd (Shanghai, China). Tetrabutyl titanate (TBT) was supplied by Sigma-Aldrich Chemical Co., Ltd (St Louis, USA). Commercial P25 was obtained from Evonik Degussa Co., Ltd (Frankfurt, Germany); it is a titania photocatalyst consisting of 20% rutile and 80% anatase, which was widely used because of its relatively high levels of activity in many photocatalytic reaction systems. The healthy zebrafishes with a body length of $(2.63 \pm 0.41)$ cm were purchased from aquarium. These zebrafishes were raised in the tank at room temperature $(25 \pm 1°C)$

and the pH value of the water was in the range of 7.2–7.5. Continuous filtration and aeration of the water were carried out throughout the day and the dissolved oxygen in water was 7.06–8.18 mg l$^{-1}$. The zebrafishes were placed under natural light for a period of 14/10 h and fed with a commercial flake food once a day. Acute toxicity experiments began after domestication for 7 days. All the reagents in this study were of analytic grade and used without further purification. Deionized water was used throughout the experiment.

## 2.2. Characterization

In order to analyse the crystalline structure of the catalysts, X-ray diffraction (XRD) was carried out on a D8 Advance diffractometer (Bruker, Germany), equipped with a Cu K$\alpha$ radiation source. All of the samples were run at Braggangles (2$\theta$) in the range of 10°–80°. The crystallinity of the samples was calculated by Jade 5.0 software. The crystallite size ($D$) was estimated by Scherrer's equation [26]

$$D = \frac{K\lambda}{B\cos\theta},$$ 

(2.1)

where $B$ is the full width at half the maximum of the XRD peak, $K$ is a constant taken at 0.94, $\theta$ is the diffraction angle and $\lambda$ was a constant taken at 1.5405 Å.

The elements and chemical states of the catalyst were confirmed by an X-ray photo-electron spectroscopy (XPS), equipped with monochromated Al-K$\alpha$ radiation (Shimadzu, Japan). The optical properties of the sample were measured by Lambda 750s UV–Vis diffuse reflectance spectra (Perkinelmer, USA), and the white barium sulfate was used as the matrix material, with a scanning wavelength in the range of 200–800 nm. The morphology was observed by an S-3000N scanning electron microscope (SEM) (Hitachi, Japan). An F-4600 PL spectrophotometer with a Xenon discharge lamp, as an excitation source, was used to detect •OH (Hitachi, Japan), with the excitation wavelength of 315 nm. The absorbance of MB was measured by U-2910 UV–Vis spectrophotometer (Hitachi, Japan). An S-7500 inductive coupled plasma emission spectrometer was applied to test the loading amount of Tb (Shimadzu, Japan).

## 2.3. Photocatalyst preparation

The photocatalysts were prepared by a facile sol–gel method [27]. In a typical preparation, TBT and absolute ethanol, named solution A, were added to the three-necked flask at room temperature and stirred continuously for 0.5 h. Meanwhile, terbium(III) nitrate hexahydrate, deionized water, glacial acetic acid and absolute ethanol, named solution B, were poured into a beaker and mixed for 0.5 h, and it was transferred to a separatory funnel and slowly added to solution A. Thereafter, the temperature was raised to 70°C, and a stable and uniform TiO$_2$ sol was obtained after 2 h of reaction. After ageing for 72 h at room temperature, the obtained wet gel was dried in an oven at 100°C. Finally, it was placed in a muffle furnace and calcined at 500°C for 4 h. The volume ratio of deionized water, TBT, absolute ethanol and glacial acetic acid was 20 : 20 : 50 : 10. The rare earth Tb doping ratio was set to 0, 0.5, 1.0, 1.5 and 2.0% according to the mass fraction of Tb to TiO$_2$.

## 2.4. Photocatalytic evaluation

The photocatalytic performance of the samples was investigated by using a 100 mg l$^{-1}$ MB solution (as the contaminant) and a high-pressure mercury lamp (125 W, 365 nm) (as the light source). Commercial P25 was used as a reference under the same experimental conditions. In a typical experiment, 100 ml of MB solution was placed in a beaker and the pH value of the solution was adjusted to a certain value. Then, 0.1 g of catalysts were added to the solution. Prior to the photocatalytic reaction, the suspension was stirred for 0.5 h under dark conditions in order to achieve an adsorption–desorption equilibrium between the organic dye molecules and the catalysts [28]. Thereafter, turning on the mercury lamp to carry out the photocatalytic reaction, 5 ml of the suspension was collected every 0.5 h. Finally, the suspension was centrifuged to remove the catalysts, and the supernatant was taken to measure the absorbance at 662 nm by a UV–visible spectrophotometer. The degradation rate was calculated by the following formula [29]

$$\eta = \frac{C_0 - C_t}{C_0} \times 100\%.$$ 

(2.2)

**Table 1.** Parameters of the samples.

| samples | exact doping amount of Tb (%) | crystallinity (%) | crystal size (nm) |
|---|---|---|---|
| pure $TiO_2$ | 0 | 99.69 | 12.7 |
| 0.5%Tb–$TiO_2$ | 0.4996 | 98.76 | 11.8 |
| 1.0%Tb–$TiO_2$ | 0.9899 | 99.18 | 10.8 |
| 1.5%Tb–$TiO_2$ | 1.4972 | 95.31 | 10.4 |
| 2.0%Tb–$TiO_2$ | 2.0046 | 97.51 | 10.1 |

where $C_0$ was the concentration of the original solution and $C_t$ was the concentration of solution at specific time intervals.

## 2.5. Acute toxicity experiment

The acute toxicity experiment was carried out by a static method [30]. Firstly, 5 l of Tb–$TiO_2$ test solution with different concentrations (12.5, 25 and 50 mg l$^{-1}$) was placed in test aquariums, and a blank control group was added only with water. Zebrafishes with a certain amount of 10 were introduced into every aquarium. During the experiment, Tb–$TiO_2$ solution was not replaced and zebrafishes were kept without feeding. The symptoms and mortality of zebrafishes at 2, 6, 24, 48, 72 and 96 h were observed and recorded. Note that the dead zebrafishes were moved in a timely manner. The $LC_{50}$ and 95% confidence intervals were calculated at 24, 48, 72 and 96 h.

## 2.6. Detection of •OH and active species trapping experiments

Hydroxyl radicals (•OH) generated by the catalysts during photocatalysis were detected by PL spectroscopy, using terephthalic acid (TPA) as the scavenger. TPA tended to react with •OH to form 2-hydroxyterephthalic acid, which is a compound with high fluorescence properties [31]. The PL intensity was in proportion to the amount of the produced •OH. In a typical procedure, $2 \times 10^{-4}$ M TPA and $1 \times 10^{-3}$ M NaOH were dissolved in 100 ml of deionized water, and MB solution was replaced by the obtained TPA solution. The other operation was the same as the photocatalytic evaluation process. In addition, $2 \times 10^{-3}$ M of *p*-benzoquinone (BQ), silver nitrate (AgNO$_3$), triethanolamine (TEOA) and isopropyl alcohol(IPA) were used as the scavengers for superoxide radicals (•O$_2^-$), electrons (e$^-$), holes (h$^+$) and •OH, respectively, for investigating the effect of active species on the photocatalytic process.

# 3. Results and discussion

## 3.1. Photocatalyst characterization

The exact Tb doping amount of the catalysts was measured by ICP. As displayed in table 1, they were basically consistent with the theoretical ones, which laid the foundation for the correct inference of the subsequent experimental rules.

In general, the photocatalytic performance of anatase was better than that of rutile; therefore, we regulated the calcined temperature at 500°C to obtain anatase [32]. The XRD patterns of the composites with different doping amounts of Tb are illustrated in figure 1. According to the peak index of Jade 5.0 software, peaks at $2\theta$ equal to 25.31°, 37.90°, 48.15°, 53.90°, 55.14° and 62.77° were indexed as (101), (004), (200), (105), (211) and (204) diffractions of anatase (PDFNO.65-5714), respectively. No diffraction peak of rutile was detected in the composites, indicating that the crystal form of $TiO_2$ did not change after Tb doping. It was worth noting that the peak intensity of Tb–$TiO_2$ was relatively weaker than that of pure $TiO_2$, and no new diffraction peak was observed due to the low doping amount of Tb. The crystallinity and crystallite size of the samples are shown in table 1. It illustrated that all the samples with a good crystallinity above 95% were obtained under a calcined temperature of 500°C. Additionally, the crystallite size of Tb–$TiO_2$ was smaller than that of pure $TiO_2$ (12.7 nm), and gradually decreased with the increase in Tb doping amount. The possible reason was that a small amount of Tb existed between the grain surface and the gap, resulting in an increase in intergranular energy barrier, thus the direct contact between crystals was hindered [33]. Besides, the force generated by the crystal distortion

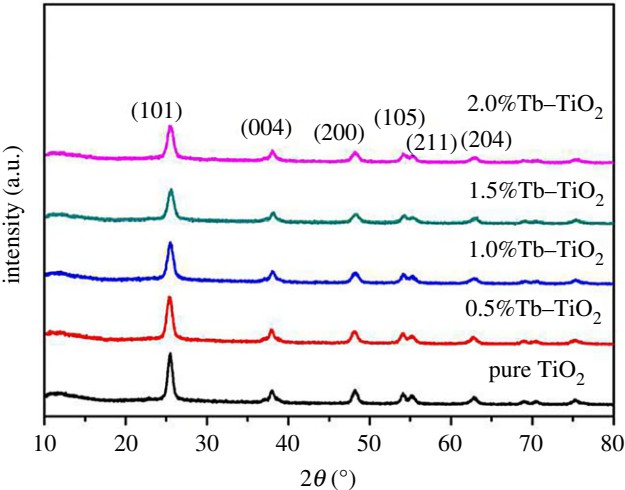

**Figure 1.** XRD patterns of Tb–TiO$_2$ samples with different Tb doping amounts.

inhibited grain growth and achieved a refinement for grain. In summary, the lattice defects of doped sample were beneficial to capture electrons. The recombination possibility of photo-generated carrier was reduced with the decrease in grain size and the increase in specific surface area. These combined factors effectively improved the photocatalytic activity.

The composition and chemical state of the elements were analysed by XPS, as presented in figure 2. Peaks of O 1s, Ti 2s, Ti 2p, Ti 3s, Ti 3p and C 1s were observed at the survey spectra (figure 2a) within the range of 1200–0 eV, indicating that the samples were mainly composed of Ti, O. As for the peak of C 1s, this could be attributed to spectrum calibration. The spectra of Tb–TiO$_2$ with asymmetric O 1 s core level are shown in figure 2b, which were fitted by two Gaussian peaks at about 530.23 eV (Ti–O–Tb) and 528.49 eV (Tb–O). It suggested that the doped sample was mainly formed by the connection of Ti–O–Tb bonds; however, there were still some terbium ions distributed on the surface of TiO$_2$, resulting in a larger oxygen content on the surface [34]. In the Ti 2p region (figure 2c), it could be fitted into four peaks in the spectrum around 456.02, 456.31, 461.78 and 462.25 eV. The peaks detected at 462.25 and 456.31 eV were assigned to Ti$^{4+}$2p$_{1/2}$ and Ti$^{4+}$2p$_{3/2}$, while the ones at about 456.02 and 461.78 eV were attributed to Ti$^{3+}$2p$_{1/2}$ and Ti$^{3+}$2p$_{3/2}$, respectively, which were formed by the splitting of the spin–orbit coupling. It confirmed the presence of Ti$^{4+}$ and Ti$^{3+}$ in the samples, where the formation of Ti$^{3+}$ was attributed to the reduction of Ti$^{4+}$ by thiourea [35], following the reaction equation $3Ti^{4+} + (H_2N)_2CS \xrightarrow{H^+} 3Ti^{3+} + [HN=C(NH_2)S]_2$. The existence of Ti$^{3+}$ could effectively restrain the recombination of photo-induced electrons and holes, and the photocatalytic activity was significantly improved. Figure 2d shows the Tb 3d spectra with a doublet, whose binding energies were at 1274.61 and 1239.76 eV, corresponding to Tb$^{3+}$3d$_{3/2}$ and Tb$^{3+}$3d$_{5/2}$ lines, respectively. The presence of Tb$^{3+}$ allowed the catalysts to have more lattice defects, which could facilitate the capture of electrons, thus, the recombination probability of photo-induced electrons and holes reduced.

Figure 3 displays the SEM images of Tb–TiO$_2$ calcined at 500°C for 4 h. As shown in figure 3a, the irregularly shaped agglomerate particle surfaces were observed. Magnification was carried out for further observation. It can clearly be seen from figure 3b that the rough surface structures were composed of uniform spherical particles with diameters of about 20 nm.

UV–visible diffuse reflectance spectroscopy (UV–vis DRS) of the samples with different Tb doping amounts is presented in figure 4. It clearly displayed that absorption bands of all the doped samples are blue-shifted towards shorter wavelengths compared to pure TiO$_2$, which could be ascribed to the well-known quantum size effect of nanomaterials [36]. The reduction in particle size resulted in a broadening of the forbidden band, so that the absorption band moved to a short wavelength. Therefore, the difference between conduction band potential and valence band potential was widened, and the redox ability of photo-induced holes and electrons was enhanced. As a result, the photo-generated carriers were easy to migrate to the surface, and the recombination in the bulk phase was reduced, leading to an enhancement of the photocatalytic activity. In addition, the absorption of Tb–TiO$_2$ at a wavelength of 380 nm was stronger than that of pure TiO$_2$, and the composite with Tb doping amount of 1.0% exhibited the strongest absorption ability. This can probably be attributed to the abundant surface state of the nanoparticles after Tb doping, and the absorption capacity of light was enhanced.

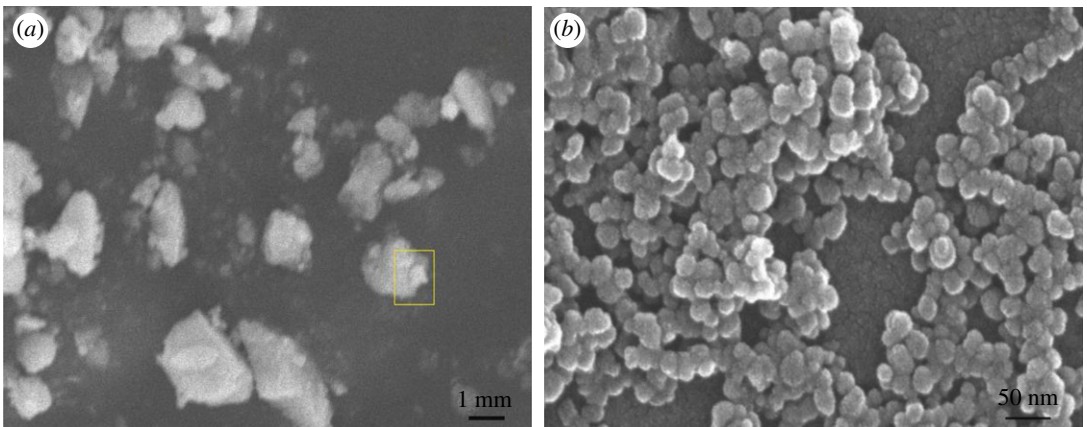

**Figure 2.** XPS patterns of 1.0%Tb–TiO$_2$ calcined at 500°C for 4 h. (*a*) Survey spectra; (*b*) O 1s region; (*c*) Ti 2p region; (*d*) Tb 3d region.

**Figure 3.** (*a*) SEM image of 1.0%Tb–TiO$_2$ composite and (*b*) a magnified image of the highlighted portion in (*a*).

## 3.2. Effect of Tb doping amount on photocatalytic activity

We investigated the effect of Tb doping on photocatalytic degradation without pH adjustment. As shown in figure 5, the photocatalytic activity first increased and then decreased as the Tb doping amount from 0 to 2.0%, and the sample with Tb doping amount of 1.0% exhibited the highest photocatalytic activity, which was consistent with the results of UV–Vis DRS. In most cases, the efficiency enhanced with the

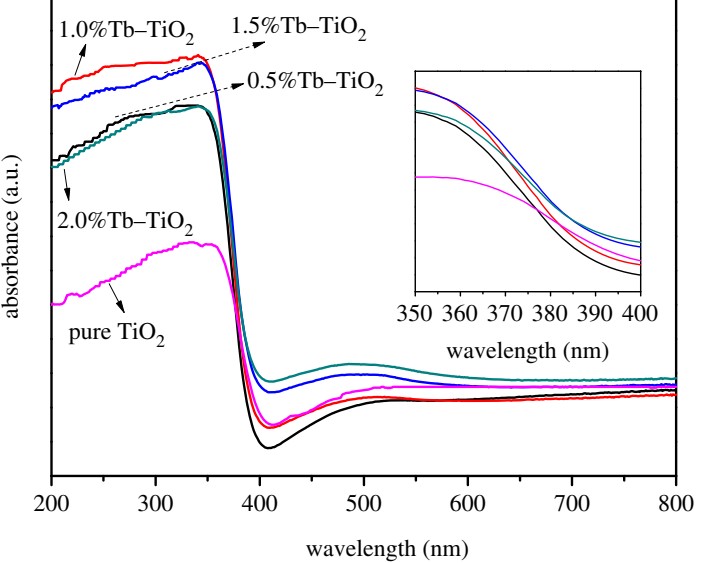

**Figure 4.** UV–Vis DRS spectra of samples with different Tb doping amounts.

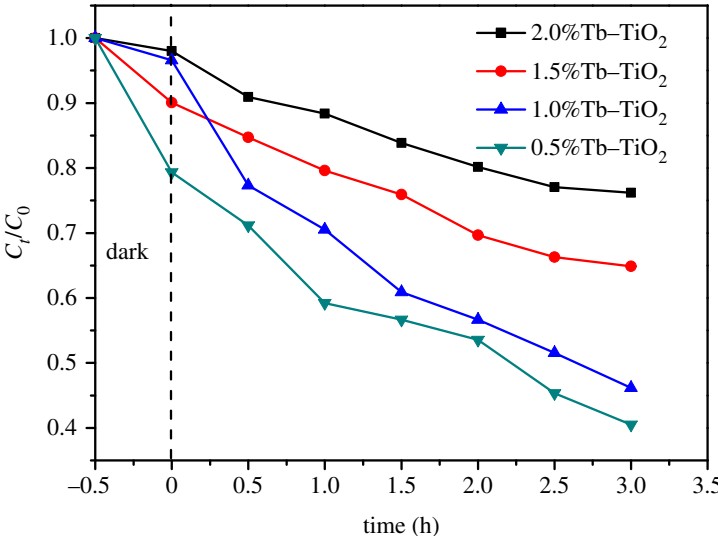

**Figure 5.** Effect of Tb doping amount on photocatalytic activity (10 mg l$^{-1}$ MB, 1.0 g l$^{-1}$ Tb–TiO$_2$ dose, $T$ = 298 K).

increase in Tb doping amount, owing to sufficient photo-induced carrier trapping centre generated in the catalyst lattice surface layer [29]. However, if the Tb doping amount was too high, a large number of defects and excessive oxygen vacancies caused by lattice distortion would result in the recombination of holes and electrons. Appropriate Tb doping amount could provide shallow traps of photo-induced electrons, which facilitated the transfer of carrier and efficient separation of charges, thereby the photocatalytic activity was improved. Therefore, the catalyst with the best photocatalytic performance (1.0%Tb–TiO$_2$) was chosen for further studies.

## 3.3. Effect of initial pH value on photocatalytic activity

It has been reported that the initial pH value of the solution is an important factor which affects the photocatalytic performance. In general, the initial pH value affects the adsorption and dissociation behaviour of reactant molecules, the surface charge of the titanium dioxide, oxidation potential and other physico-chemical properties [37]. Besides, electrostatic attraction or repulsion between the catalysts and the reactant molecules results in an enhancement or suppression of the photocatalytic rate. Accordingly, the effect of the initial pH value on photocatalytic activity was studied in the range

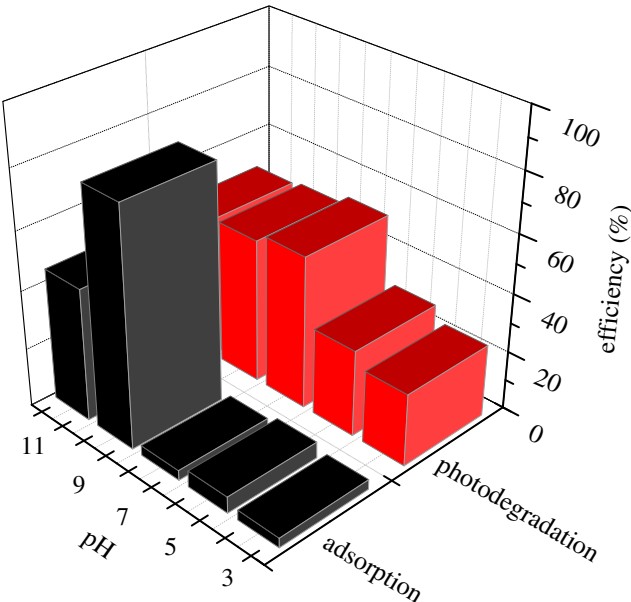

**Figure 6.** Effect of initial pH value on photocatalytic activity (10 mg l$^{-1}$ MB, 1.0 g l$^{-1}$ Tb–TiO$_2$ dose, $T = 298$ K).

of 3–11, using hydrochloric acid and sodium hydroxide as the adjust reagents. As illustrated in figure 6, the adsorption capability was kept substantially constant when the initial pH value from 3 to 7. With the further increase in pH value, adsorption capability was greatly increased, exhibiting the highest capability of 74% when the initial pH value was 9. Generally, acidic conditions help with the formation of hydrogen bonds, but too strong acidity would destroy the structure of the catalysts [38]. On the contrary, when the pH value was too high (pH > 9), a competitive adsorption between MB molecules and excessive OH$^-$ ions could exist. Besides, the occurrence of a deprotonation process would reduce positive charge on the surface of Tb–TiO$_2$, leading to a poor adsorption capacity in the subsequent photocatalysis. As for the photocatalytic efficiency, it was first increased and then decreased with the increase in pH value, and exhibited the highest photocatalytic efficiency when the initial pH value was 7. The effect of initial pH value on the photocatalysis and adsorption of MB was not completely consistent. This interesting phenomenon indicated that the photocatalytic activity of Tb–TiO$_2$ composites depended on the combined effect of multiple factors, in addition to its adsorption capacity. Note that the general pH value in dye wastewater is 6–8, demonstrating the Tb–TiO$_2$ composites were suitable for wastewater treatment in the textile industry without preliminary pH adjustment. Therefore, it is unnecessary to adjust pH in the subsequent experiments.

## 3.4. The maximum degradation capability of MB

In order to investigate the maximum degradation capability of MB, several distinct treatments were carried out towards the experimental system: (i) the photolysis without any catalysts operating only with UV light irradiation; (ii) the adsorption without light irradiation, in the presence of Tb–TiO$_2$; (iii) the photocatalysis under UV light irradiation with P25 and Tb–TiO$_2$. As shown in figure 7$a$, the ordinate represents the ratio of the MB concentration at a certain time to the MB concentration in the initial solution. The results indicated that photolysis and adsorption process had little effect on the degradation of MB. On the contrary, photocatalysis under UV light irradiation with P25 and Tb–TiO$_2$ showed better degradation capability to MB, and Tb–TiO$_2$ achieved an excellent photocatalytic efficiency of 54.3% in 3 h, which was higher than that of the P25.

Figure 7$b$ presents the fitting curves of pseudo-first-order kinetics. It clearly displays that $-\ln(C_0/C_t)$ were linearly related to $t$, and the correlation coefficients were greater than 97%, indicating that the results of photocatalysis with Tb–TiO$_2$ and P25 were fitted with the following pseudo-first-order equation. Tb–TiO$_2$ exhibited approximately three times enhancement in photocatalytic activity with a reaction rate constant of 0.2314 h$^{-1}$ when compared with that of commercial P25 (0.0827 h$^{-1}$). UV–vis spectra of MB degraded at a given time are shown in figure 8. It was observed that the characteristic adsorption peaks of MB at 662 nm weakened along with the increase in irradiation time. The corresponding

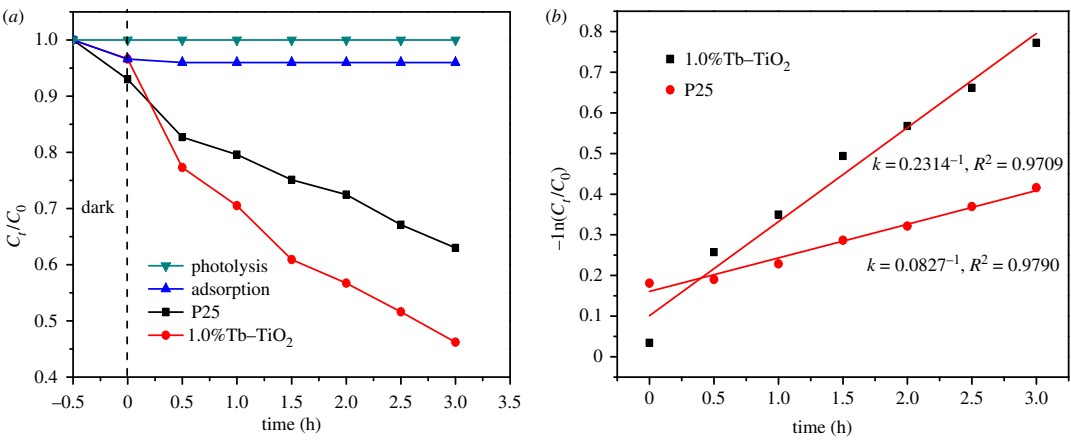

**Figure 7.** (*a*) Comparison of photocatalytic activity with different treatments; (*b*) fitting curve of pseudo-first-order kinetics.

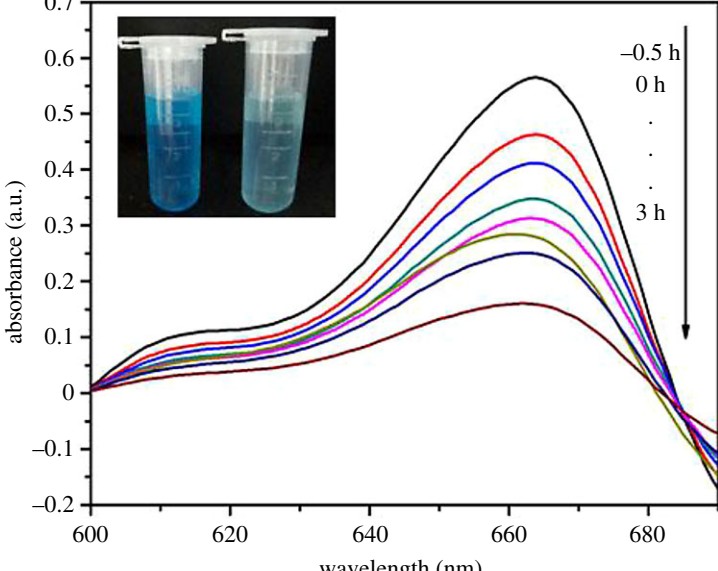

**Figure 8.** UV–Vis spectra of MB degraded at a given time.

photographs displayed that the colour of MB solution was changed from deep blue to light blue after photocatalysis (see inset of figure 8). This phenomenon indicated that the MB molecules could be efficiently degraded by Tb–TiO$_2$ [39].

## 3.5. Catalyst reusability

The reusability is an important parameter to measure the potential application value of the catalysts. In this work, recycling experiments were applied to investigate the reusability of the catalysts [27]. After the first cycle of photocatalysis, the catalysts were collected by a filtration process and washed with ethanol. Thereafter, the used catalysts were employed to degrade a fresh 10 mg l$^{-1}$ MB solution under the same conditions. This operation was repeated three times. Finally, the catalysts were collected after the third cycle, and the catalysts were calcined and regenerated at 500°C for use in the fourth cycle. As shown in figure 9, a slight decrease (4.3%) in degradation efficiency from the first to the third cycle was observed. It was probably caused by the adsorption of organic intermediates on the catalyst surface, which inhibited the rapid diffusion of the organic dye molecules to the surface or interface of the catalysts, thereby the subsequent photocatalytic reaction was hindered. Note that the photocatalytic activity of the used catalysts in the fourth cycle (51.3%) was basically the same as the fresh ones (52.2%), indicating that the calcination was beneficial to the regeneration of the catalysts. It has been confirmed that the resulting catalysts were reusable for at least four cycles without significant loss of activity.

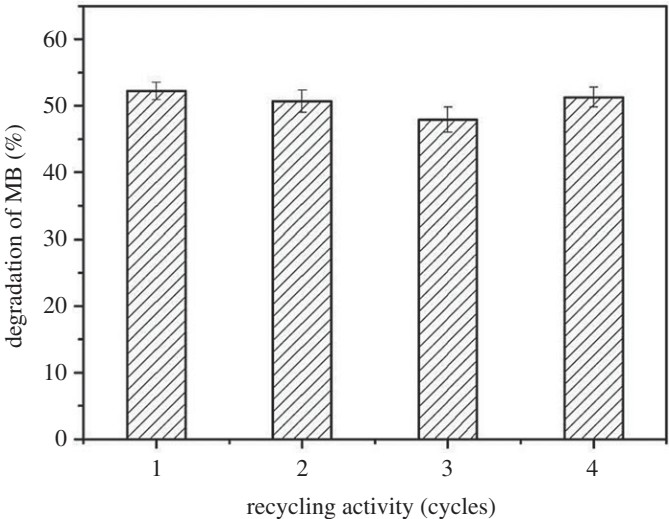

**Figure 9.** The photocatalytic activity of 1.0%Tb–TiO$_2$ at different periodic cycles.

**Table 2.** Results of zebrafishes' acute toxicity tested by Tb–TiO$_2$. A, normal; B, swimming retardation; I, breath weakness; O, afloat; P, all death.

| catalyst concentration (mg l$^{-1}$) | 2 h | 6 h | 24 h | 48 h | 72 h | 96 h |
|---|---|---|---|---|---|---|
| 0 | A | A | A | A | A | A |
| 12.5 | A | A | A | A | A | O, B, I |
| 25 | A | A | A | O, B, I | O, B, I | P |
| 50 | A | O, B, I | O, B, I | P | P | P |
| LC$_{50}$ (mg l$^{-1}$) | — | — | 41.6 | 32.8 | 27.3 | 23.2 |
| 95% confidence interval (mg l$^{-1}$) | — | — | 39.6–43.6 | 31.1–34.5 | 25.5–28.1 | 21.9–24.5 |

## 3.6. Acute toxicity of catalysts on zebrafishes

To investigate the effects of Tb–TiO$_2$ on the aquatic ecosystem, we evaluated its ecological safety through acute toxicity experiments, and the results are displayed in table 2 and figure 10. Based on the concentration of the test solution and the mortality of zebrafishes, the acute toxicity of Tb–TiO$_2$ on zebrafishes was 23.2 mg l$^{-1}$ (96 h-LC$_{50}$). According to the toxicity grading standards of China, Tb–TiO$_2$ exhibited a low toxicity on zebrafishes (LC$_{50}$ > 10 mg l$^{-1}$) [40].

## 3.7. Photocatalytic mechanism discussion

The doping of Tb could provide more lattice defects, which acted as shallow traps of photo-induced electrons, facilitating the transfer of carrier and efficient separation of charges. Besides, the formation of Ti–O–Tb bond provided more surface defects and broke the original equilibrium to form Lewis acid sites. On the one hand, these surface defects have been reported to not only have the ability to capture photo-generated carriers, but also to increase the reactivity of hydroxyl radicals. On the other hand, an increment of surface acidity provided more adsorption sites. These combined factors effectively improved the photocatalytic activity.

It is generally considered that a series of active species are produced by the catalysts during photocatalysis, including photo-induced holes (h$^+$), hydroxyl radicals (•OH) and superoxide radicals (•O$_2^-$) [41]. In order to reveal the photocatalytic mechanism of Tb–TiO$_2$ composites for degrading MB, fluorescence spectrophotometry was used to trace the •OH generated at a given time during the photocatalytic process. In addition, triethanolamine (TEOA), silver nitrate (AgNO$_3$), $p$-benzoquinone

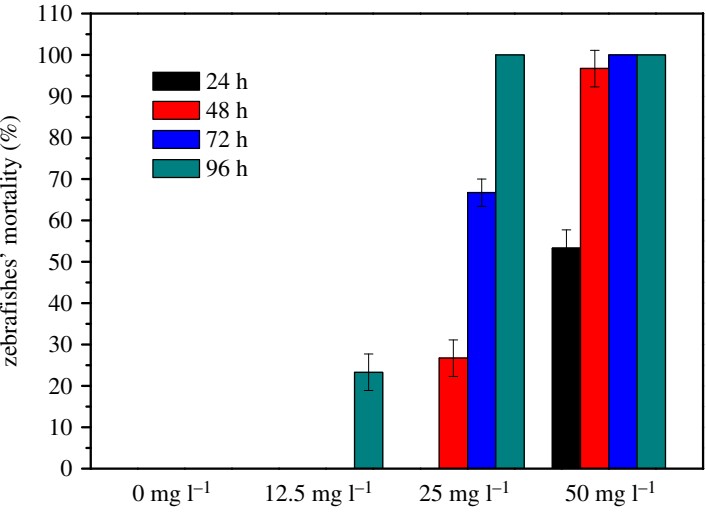

**Figure 10.** Mortality of zebrafishes in the solution with different Tb–TiO$_2$ concentrations.

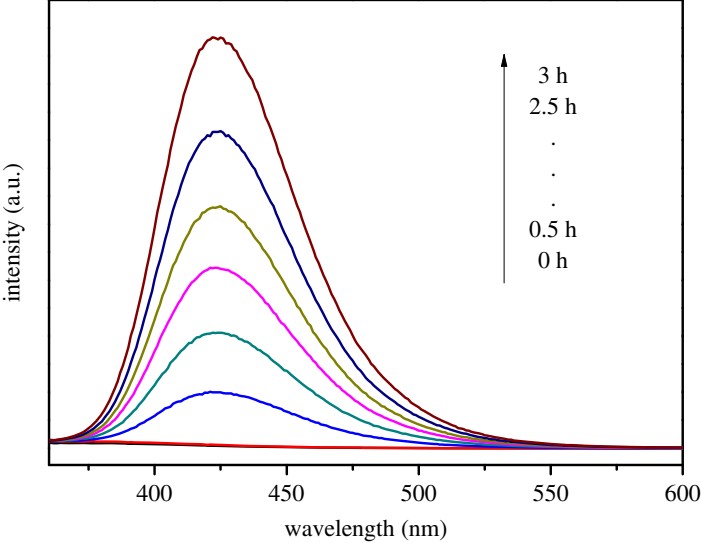

**Figure 11.** PL spectra of ·OH trapping on TA solution under UV light irradiation.

(BQ) and isopropanol (IPA) were used as the scavengers of h$^+$, e$^-$, •O$_2^-$, •OH, respectively, to investigate the effect of different active species on photocatalytic degradation of MB.

•OH trapping PL spectra are shown in figure 11. The obvious enhancement of fluorescence intensity at 425 nm was observed along with the increase in irradiation time, which can be attributed to the production of •OH. The PL intensity increased evenly every 0.5 h, which was not completely consistent with the results of photocatalytic experiments, indicating that •OH radical was not the only dominant active species in the photocatalytic oxidation system. The active species trapping experiments are displayed in figure 12, when TEOA (h$^+$) or IPA (•OH) was added into the photocatalytic reaction system, the photocatalytic efficiency of MB was obviously suppressed. The reaction rate constants were of 0.0720 h$^{-1}$ and 0.0429 h$^{-1}$, respectively, which were significantly less than the reaction rate without any scavenger. On the contrary, the addition of AgNO$_3$ (e$^-$) and BQ (•O$_2^-$) showed a slighter effect on it. The above results indicated that h$^+$ and •OH were the dominant active species for MB photocatalytic process in the presence of Tb–TiO$_2$, and •O$_2^-$ and e$^-$ played a relatively minor role.

Based on the above results and analysis, we proposed a possible photocatalytic mechanism of MB degraded by Tb–TiO$_2$. As shown in figure 13, the holes (h$^+$) and free electrons were generated by Tb–TiO$_2$ under the excitation of UV light. The free electrons reacted with surface defects of Tb–TiO$_2$, resulting in the neutralization of some electrons, and thereby excess holes (h$^+$) were generated.

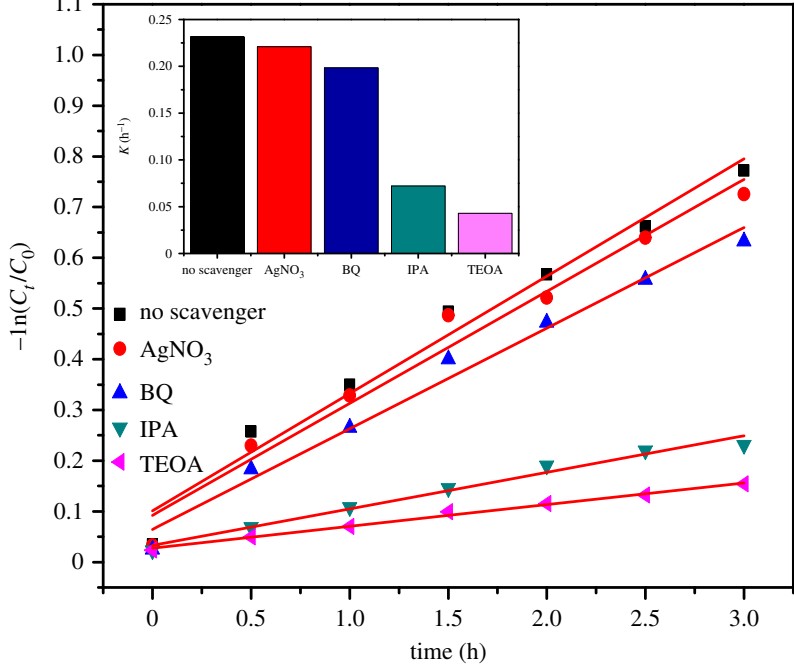

**Figure 12.** The photocatalytic degradation of MB with different scavengers.

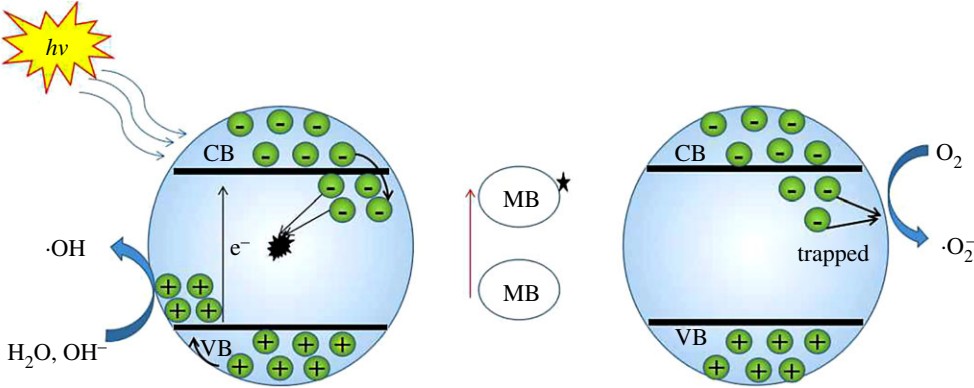

**Figure 13.** The photocatalytic mechanism of MB degraded by Tb–TiO$_2$ under UV light irradiation.

Subsequently, the excess holes (h$^+$) reacted with H$_2$O and OH$^-$ on the catalyst surface to produce •OH. When MB was added to the system, it was easily adsorbed on the surface of Tb–TiO$_2$, and its electrons were transferred to the conduction band of Tb–TiO$_2$, thus the electron of Tb–TiO$_2$ conduction band was getting abundant and O$_2$ on the catalyst surface was reduced to •O$_2^-$ by electrons. At this time, both •OH and •O$_2^-$ entered into the solution, thereby MB was degraded into small molecules.

# 4. Conclusion

The Tb–TiO$_2$ composites were successfully prepared by a sol–gel method with a facile step. The amount of Tb doping and the initial pH value of the solution had a great influence on the photocatalytic efficiency. The resulting 1%Tb–TiO$_2$ composite with a high crystallinity of 99.18% and an average crystallite size of 10.8 nm exhibited the best photocatalytic performance for MB without pH adjustment. It showed approximately three times enhancement in photocatalytic activity with a reaction rate constant of 0.2314 h$^{-1}$ when compared with that of commercial P25 (0.0827 h$^{-1}$), owing to the contributions of appropriate amounts of lattice defects formed in the composites, which could facilitate the capture of electrons, thus the recombination probability of photo-induced electrons and holes reduced. The photocatalytic process of MB was suitable for the Langmuir model, and it

followed the pseudo-first-order kinetic. The catalysts presented low toxicity on zebrafishes with 96 h-$LC_{50}$ of 23.2 mg l$^{-1}$ and have been proved to be reusable for at least four cycles without significant loss of activity. Active species trapping experiments demonstrated that h$^+$ and •OH were the dominant active species in this photocatalytic process. As a highly efficient photocatalyst for MB, Tb–$TiO_2$ composite could be a promising candidate to degrade organic pollutants from aqueous solutions in environmental pollution management in the near future.

Data accessibility. The raw data are deposited at the Royal Society's figshare portal (https://rs.figshare.com/) free of charge.

Authors' contributions. Z.W. and Y.S. participated in the design of the study and drafted the manuscript; X.C. and J.Z. collected and analysed the data; T.T. provided a lot of advice and guidance for the analysis of sample optical properties and English improvement; S.W. conceived, designed and coordinated the study and helped draft the manuscript. All authors gave final approval for publication and agreed to be held accountable for the work performed therein.

Competing interests. We declare we have no competing interests.

Funding. This project was financially supported by the Hainan Province Natural Science Foundation of China (grant no. 219QN220) and the Scientific research Foundation of Hainan Medical University (grant no. HY2018-17).

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
