## [Reviewer comments · Royal Society Open Science]

Review History

RSOS-191077.R0 (Original submission)

Review form: Reviewer 1

Is the manuscript scientifically sound in its present form?

Yes

Are the interpretations and conclusions justified by the results?

Yes

Is the language acceptable?

Yes

Do you have any ethical concerns with this paper?

No

Have you any concerns about statistical analyses in this paper?

No

Recommendation?

Accept with minor revision (please list in comments)

Comments to the Author(s)

In this MS, the authors described a sol-gel route for preparing Tb-doped TiO₂ nanoparticles and investigated the photocatalytic performance to degrade MB. Since metal ions-doped TiO₂ photocatalysts have been widely reported in the literature, the present work has not enough novelty from the synthetic methodology. Nevertheless, some findings are interesting such as biosafety. The present MS can be considered for publication in this journal after the below queries are answered.

1. According to Table 1, with the increase of Tb content the size of the product decreases. However, the crystallinity is irregular. Why?
2. The authors claimed that the presence of C peak was attributed to exogenous pollution. This is unclear description. In fact, the C peak is always detected in XPS survey spectrum. Is it polluted at all times? Moreover, the formation of Ti³⁺ was also attributed to the reduction of Ti⁴⁺. Then, please give related reaction equations.
3. In P3, Line 11: The authors should cite more references on metal ions-doped TiO₂ catalysts, such as J. Colloid Interface Sci., 2007, 315, 382; Mater. Lett., 2008, 62, 895; Dalton Transactions, 2011, 40, 3689; and so on.
4. The present expression of superoxide radicals is error. Please correct it.
5. English should be carefully checked to avoid many errors/mistakes.

Review form: Reviewer 2

Is the manuscript scientifically sound in its present form?

Yes

Are the interpretations and conclusions justified by the results?

Yes

Is the language acceptable?

Yes

Do you have any ethical concerns with this paper?

No

Have you any concerns about statistical analyses in this paper?

No

Recommendation?

Major revision is needed (please make suggestions in comments)

Comments to the Author(s)

The manuscript reported the preparation and performance of Tb doped TiO₂. I think it showed some interest and can be accepted after some revisions.

1. Fig. 1 is suggested to be deleted.
2. Why the absorption of the samples are below zero in the region of 200-250 nm?
3. Why 0.5%Tb-TiO₂ exhibited the highest adsorption performance?
4. More discussion should be supplemented on the mechanism of enhanced activity.

5. What are the exact amount of Tb in the doped samples?
6. Some recent references on the doped TiO₂ are suggested to be cited. Chemical Engineering Journal, 2018, 352, 947-956; J. Mater. Chem. A, 2016, 4, 8682-8689; J. Phys. Chem. C, 2012, 116, 17721-17728; Journal of hazardous materials, 2018, 351, 196-205;

Decision letter (RSOS-191077.R0)

22-Jul-2019

Dear Dr Wen:

Title: Rapid Preparation of Terbium Doped Titanium Dioxide Nanoparticles and Its Enhanced Photocatalytic Performance
Manuscript ID: RSOS-191077

The editor assigned to your manuscript has now received comments from reviewers. We would like you to revise your paper in accordance with the referee and Subject Editor suggestions which can be found below (not including confidential reports to the Editor). Please note this decision does not guarantee eventual acceptance.

Please submit your revised paper before 14-Aug-2019. Please note that the revision deadline will expire at 00.00am on this date. If we do not hear from you within this time then it will be assumed that the paper has been withdrawn. In exceptional circumstances, extensions may be possible if agreed with the Editorial Office in advance. We do not allow multiple rounds of revision so we urge you to make every effort to fully address all of the comments at this stage. If deemed necessary by the Editors, your manuscript will be sent back to one or more of the original reviewers for assessment. If the original reviewers are not available we may invite new reviewers.

Please also include the following statements alongside the other end statements. As we cannot publish your manuscript without these end statements included, if you feel that a given heading is not relevant to your paper, please nevertheless include the heading and explicitly state that it is not relevant to your work.

- Acknowledgements

- Funding statement

Please include a funding section after your main text which lists the source of funding for each author.

RSC Associate Editor:
Comments to the Author:
(There are no comments.)

RSC Subject Editor:
Comments to the Author:
(There are no comments.)

Reviewers' Comments to Author:
Reviewer: 1

Comments to the Author(s)

In this MS, the authors described a sol-gel route for preparing Tb-doped TiO₂ nanoparticles and investigated the photocatalytic performance to degrade MB. Since metal ions-doped TiO₂ photocatalysts have been widely reported in the literature, the present work has no enough novelty from the synthetic methodology. Nevertheless, some findings are interesting such as biosafety. The present MS can be considered for publication in this journal after the below queries are answered.

1. According to Table 1, with the increase of Tb content the size of the product decreases. However, the crystallinity is irregular. Why?
2. The authors claimed that the presence of C peak was attributed to exogenous pollution. This is unclear description. In fact, the C peak is always detected in XPS survey spectrum. Is it polluted at all times? Moreover, the formation of Ti³⁺ was also attributed to the reduction of Ti⁴⁺. Then, please give related reaction equations.
3. In P3, Line 11: The authors should cite more references on metal ions-doped TiO₂ catalysts, such as J. Colloid Interface Sci., 2007, 315, 382; Mater. Lett., 2008, 62, 895; Dalton Transactions, 2011, 40, 3689; and so on.

4. The present expression of superoxide radicals is error. Please correct it.
5. English should be carefully checked to avoid many errors/mistakes.

Reviewer: 2

Comments to the Author(s)

The manuscript reported the preparation and performance of Tb doped TiO₂. I think it showed some interest and can be accepted after some revisions.

1. Fig. 1 is suggested to be deleted.
2. Why the absorption of the samples are below zero in the region of 200-250 nm?
3. Why 0.5%Tb-TiO₂ exhibited the highest adsorption performance?
4. More discussion should be supplemented on the mechanism of enhanced activity.
5. What are the exact amount of Tb in the doped samples?
6. Some recent references on the doped TiO₂ are suggested to be cited. Chemical Engineering Journal, 2018, 352, 947-956; J. Mater. Chem. A, 2016, 4, 8682-8689; J. Phys. Chem. C, 2012, 116, 17721-17728; Journal of hazardous materials, 2018, 351, 196-

Author's Response to Decision Letter for (RSOS-191077.R0)

See Appendix A.

RSOS-191077.R1 (Revision)

Review form: Reviewer 1

Is the manuscript scientifically sound in its present form?

Yes

Are the interpretations and conclusions justified by the results?

Yes

Is the language acceptable?

Yes

Do you have any ethical concerns with this paper?

No

Have you any concerns about statistical analyses in this paper?

No

Recommendation?

Accept as is

Comments to the Author(s)

The present revised MS can be accepted.

Decision letter (RSOS-191077.R1)

02-Sep-2019

Dear Dr Wen:

Title: Rapid Preparation of Terbium Doped Titanium Dioxide Nanoparticles and Its Enhanced Photocatalytic Performance
Manuscript ID: RSOS-191077.R1

It is a pleasure to accept your manuscript in its current form for publication in Royal Society Open Science. The chemistry content of Royal Society Open Science is published in collaboration with the Royal Society of Chemistry.

RSC Associate Editor:
Comments to the Author:
(There are no comments.)

RSC Subject Editor:
Comments to the Author:
(There are no comments.)

Reviewer(s)' Comments to Author:
Reviewer: 1

Comments to the Author(s)
The present revised MS can be accepted.

Appendix A

Response to referees

Dear Editors and Reviewers,

Thank you for your letter and for the reviewers' comments concerning our manuscript entitled "Rapid Preparation of Terbium Doped Titanium Dioxide Nanoparticles and Its Enhanced Photocatalytic Performance" (ID: RSOS-191077). Those comments are all valuable and very helpful for revising and improving our paper, as well as the important guiding significance to our researches. We have studied comments carefully and have made correction which we hope to meet with approval. Revised portion are marked in red in the paper. The main corrections in the paper and the responds to the reviewer's comments are as flowing:

Reviewer: 1

1. According to Table 1, with the increase of Tb content the size of the product decreases. However, the crystallinity is irregular. Why?

Response: The crystallinity is related to the pH value of the reaction solution and the amount of Tb doped into the titanium dioxide lattice, in addition to the calcined process. However, since acetic acid was added during the preparation of samples, the pH value of the reaction solution varied dynamically depending on the amount of terbium(III) nitrate hexahydrate added. Besides, the amount of terbium doped into the titanium dioxide lattice or the surface was uncontrollable. In summary, due to the complexity of these two factors, the crystallinity of all samples was higher than 95%, but it didn't show obvious regularity.

2. The authors claimed that the presence of C peak was attributed to exogenous pollution. This is unclear description. In fact, the C peak is always detected in XPS survey spectrum. Is it polluted at all times? Moreover, the formation of Ti^{3+} was also attributed to the reduction of Ti^{4+} . Then, please give related reaction equations.

Response: We are very sorry that we have made a mistake for the explanation of C

peak detected in XPS survey spectrum. In fact, the presence of C peak was attributed to spectrum calibration. It has been corrected in part 3.1. Moreover, Ti^{4+} was reduced to Ti^{3+} by thiourea under acidic conditions, the related reaction equation was displayed in in part 3.1.

3. In P3, Line 11: The authors should cite more references on metal ions-doped TiO_2 catalysts, such as J. Colloid Interface Sci., 2007, 315, 382; Mater. Lett., 2008, 62, 895; Dalton Transactions, 2011, 40, 3689; and so on.

Response: The literature mentioned above has been cited in the revised version, marked as [13-15].

4. The present expression of superoxide radicals is error. Please correct it.

Response: The expression of superoxide radicals has been corrected in part 3.7.

5. English should be carefully checked to avoid many errors/mistakes.

Response: We have carefully checked the full text, and detailed corrections have been marked in red.

Reviewer: 2

1. Fig. 1 is suggested to be deleted.

Response: It has been deleted in the revised version.

2. Why the absorption of the samples are below zero in the region of 200-250 nm?

Response: We are very sorry that due to the instrument failure, there may be a large deviation in the results caused by the instability of the baseline.. Now we have re-tested, the spectrum is as shown in Fig. 4 of the revised draft.

3. Why 0.5%Tb- TiO_2 exhibited the highest adsorption performance?

Response: According to the report of Reszczyńska J, the specific surface area of

catalysts increases after doping rare earth oxides (except for Yb³⁺ doping) compared to undoped TiO₂. Nanomaterials with a large specific surface surface will have a significant increase in surface hydroxyl groups, leading to a remarkable increase of adsorption performance. However, too high doping concentration will saturate the doping ions in the nano TiO₂ lattice, thereby, the effective surface area of TiO₂ reduces, and cause the adsorption performance to decrease. Therefore, 0.5%Tb-TiO₂ exhibited the highest adsorption performance.

4. More discussion should be supplemented on the mechanism of enhanced activity.

Response: It has been supplemented in part 3.7.

5. What are the exact amount of Tb in the doped samples?

Response: The exact amount of Tb in the doped samples has been listed in table 1.

6. Some recent references on the doped TiO₂ are suggested to be cited. Chemical Engineering Journal, 2018, 352, 947-956; J. Mater. Chem. A, 2016, 4, 8682 - 8689; J. Phys. Chem. C, 2012, 116, 17721-17728; Journal of hazardous materials, 2018, 351, 196-205;

Response: The literature mentioned above has been cited in the revised version, marked as [21-24].